# Temporal Trends in Apparent Energy and Macronutrient Intakes in the Diet in Bangladesh: A Joinpoint Regression Analysis of the FAO’s Food Balance Sheet Data from 1961 to 2017

**DOI:** 10.3390/nu12082319

**Published:** 2020-08-02

**Authors:** Syed Mahfuz Al Hasan, Jennifer Saulam, Kanae Kanda, Akitsu Murakami, Yusuke Yamadori, Yukinori Mashima, Nlandu Roger Ngatu, Tomohiro Hirao

**Affiliations:** 1Department of Public Health, Faculty of Medicine, Kagawa University, Kagawa 7610793, Japan; oda@med.kagawa-u.ac.jp (K.K.); akitsu@med.kagawa-u.ac.jp (A.M.); yamadori@med.kagawa-u.ac.jp (Y.Y.); ymashima@med.kagawa-u.ac.jp (Y.M.); ngatu@med.kagawa-u.ac.jp (N.R.N.); sharks@med.kagawa-u.ac.jp (T.H.); 2Department of Food Processing and Nutrition, Karnataka State Akkamahadevi Women’s University, Vijayapura, Karnataka 586108, India; jennifersaulam27@gmail.com

**Keywords:** apparent energy intake, apparent macronutrient intake, food balance sheet, joinpoint regression analysis, jump model

## Abstract

We analyzed the temporal trends and significant changes in apparent energy and macronutrient intakes in the Bangladeshi diet from 1961 to 2017. Due to the lack of a long-running national dietary intake dataset, this study used the Food and Agriculture Organization (FAO)’s old and new food balance sheet dataset. We used the joinpoint regression model and jump model to analyze the temporal trends in apparent energy and macronutrient intakes. The annual percentage change (APC) was computed for each segment of the trends. Bangladesh has experienced a late energy revolution in their dietary history. During the 1960s, 1970s, 1980s, and 1990s, Bangladesh was suffering from substantive calorie deficits, where in apparent energy intake was less than 2200 kcal/day/person. Since the late 1990s, Bangladesh has made significant progress in raising the apparent energy consumption in the diet. Since the late 1970s, apparent fat intake started to increase significantly at a marked rate (APC = 2.16), whereas since the early 1990s, protein intake increased significantly by 1.33% per year. Plant sources have mostly governed the protein and fat intake trends in the Bangladeshi diet since 1960, whereas animal sources began to contribute significantly in protein intake since 1990 (APC = 3.43) and in fat intake since 2000 (APC = 2.88). Bangladesh overcame the substantive calorie deficit condition in the diet from the late 1990s. Excessive carbohydrate intake along with imbalanced and low-quality protein and fat intakes have been the central features in the diet in Bangladesh.

## 1. Introduction

Food consumption is the key variable to measure and evaluate the evolution of the worldwide food situation. The world has made considerable progress in raising the per capita dietary energy and food availability at the national level [1]. At a gross level, the nutrition transition can be defined as changes in per capita energy supplies. Today’s per capita energy availability compared to that of past centuries shows an almost universal trend toward higher availability levels [2]. This higher availability trend is accompanied by the changes in dietary history, which have quickened over the last three centuries and after the Second World War, having gained momentum [3]. The accelerated dietary changes might have been linked to the global emergence of diet-related non-communicable diseases (DR-NCDs). Developing countries are experiencing a rapid nutrition transition together with the existence of the double burden of disease [4]. Like other developing countries, Bangladesh is not an exclusion of this transition. Although Bangladesh has experienced declines in the proportion of stunted and underweight children and the percent of households with ultra and extreme poor [5], double burden of diseases have emerged as a major public health problem. In addition, about 61% of the total burden of disease in Bangladesh is accounted for by DR-NCDs, premature death, disability, and life lost due to ill health [6].

Bangladesh has made a remarkable improvement in food grain production and food availability at the national level [7], although food and nutrition security at household as well as national levels remains a big challenge for the government. At national level, cereals remained the bulkiest available food in the diet and have not changed markedly since 1961. Since the late 1970s, vegetable oil availability; since the late 1980s, animal-sourced foods (eggs, meat, and fish) availability; and the since the early 2000s, milk, vegetable, and fruit availability in the diet started to increase, however, the per capita amounts were grossly inadequate [7]. Moreover, the typical diet in Bangladesh at the individual, as well as at household level, is not balanced and remains dominated by cereals, mostly rice [8,9,10]. Dating back to the first ever and moreover the benchmark nutrition survey in 1962–1964, containing a wealth of data in Bangladesh, this research showed that a big majority of households (about 60%) had inadequate protein intake (57.5 g/day/person in rural areas and 49.5 g/day/person in urban areas) and almost half (about 46%) of them had inadequacy in energy (2251 kcal/day/person in rural areas and 1732 kcal/day/person in urban areas) [11]. Following the independence of Bangladesh in 1971, the results from the Bangladesh National Nutrition Surveys in 1962–1964, 1975–1976, 1981–1982, and 1995–1996 have shown that most of the households had inadequate energy and protein intakes in their diet [12]. In addition, per capita daily intakes of energy (2094 kcal in 1975–1976; 1943 kcal in 1981–1982), protein (58.5 g in 1975–1976; 48.4 g in 1981–1982), and fat (12.2 g in 1975–1976; 9.8 g in 1981–1982) in rural Bangladesh reduced from 1975–1976 to 1981–1982, with the quality of the protein being low and with there being a virtual absence of food from animal sources [10]. Moreover, the Household (Income and) Expenditure Survey (H(I)ES) data suggest that consumption of starches (g/day/person) stayed stable, and pulse (g/day/person) consumption decreased markedly in the diet since 1985 [8]. An acceptable rise in the intake of fish, eggs, vegetables, and spices have been reported since the 1980s [8,9]. Food intake from diverse food groups have expanded over time, yet it is grossly inadequate and far underneath the recommended level for fish (60 g/day), eggs (30 g/day), meat (40 g/day), milk (130 g/day), vegetables (300 g/day), fruits (100 g/day), and pulses (50 g/day) [8].

Tracing and monitoring diet at individual as well as community levels is required to evaluate the effect of dietary changes over time [7]. In addition, obtaining information on apparent and actual intakes is fundamental for foreseeing future dietary changes. In our prior study, we revealed the temporal trends in apparent food intakes in Bangladesh from 1961 to 2013 [7]. Other than this, only a few investigations have reported the changes in diet in Bangladesh at certain points in time because of the absence of a long-running dietary database in Bangladesh [8,9]. In Bangladesh, to contemplate the trends and dietary changes at a national level, data sources are limited, and since 1973, only the H(I)ES in the years1985, 1988, 1991, 1995, 2000, 2005, and 2010 are accessible [8]. Because of the absence of long-running, individual-level dietary data in many countries, food consumption at the national level is derived from food supply information, compiled in the food balance sheets (FBSs).

The FBSs provide a wider perspective of the pattern of food availability at a national level during a specific reference period [13]. The FBS is compiled in each year and is an international resource of a country’s food availability. Furthermore, this dataset assists in comparing comprehensive dietary trends over a very long time span and between countries. FBSs provide a reasonable tentative image of the overall food situation of a country, which is valuable for economical and nutritional studies for creating plans and figuring ventures [14]. Data on the national food availability, in terms of kcal/person/day, consumed within a nation provide worthy information and insights into diets and their consecutive assessment over time [15]. Henceforth, this study intended to analyze the temporal trends and noteworthy changes in apparent energy and macronutrient consumption in Bangladesh from 1961 to 2017. We additionally discussed some of the potential drivers that might have acted to change the apparent energy and macronutrient intakes in the diet in Bangladesh.

## 2. Methods

### 2.1. Data Sources and Compilation

Like most countries, food supply derived from FBSs is a comparatively dependable and conceivably the only database available to pursue and analyze the trends of dietary transition nationally. In this study, because of the absence of a long-running dietary intake dataset at the national level of Bangladesh, we obtained energy and nutrient availability data from the Food and Agriculture Organization (FAO)’s food balance sheets documented in the Food and Agriculture Organization Corporate Statistical Database (FAOSTAT). For the very first time, we used the old FBS data from 1961 to 2013, the latter being the last available year in the old FBS [16], and the new FBS from 2014 to 2017, the latter being the latest available year in the new food balance sheets [17]. The new FBS has some methodological advancement in computation compared to the old FBS. The key difference between the new and old FBS methodologies is the absence of a balancer variable such as stocks [18]. In the past, for example, stocks or feed components of the FBS inherited almost all the statistical errors. In the new FBS with the new methodology, dedicated modules and a balancing mechanism are used to effectively impute the FBS components and spread out the imbalances [19]. For this study, the FBSs were downloaded as two comma separated values (.csv) files from the FAOSTAT database—one was from the old FBS from 1961 to 2013 and another one was from the new FBS from 2014 to 2017. After that, we had merged the two databases and made one single database from 1961 to 2017. Per capita energy, protein, and fat availability data are represented as kcal/person per day, g/person per day, and g/person per day, respectively, in the FAOSTAT database [16,17].

### 2.2. Operational Definition

Food availability provides reliable information on the apparent intake instead of actual intake in the diet [20]. Since the data come from national food balance sheets rather than nationwide dietary survey, these intake data refer to “average energy and macronutrient available for consumption”. Hence, in the remainder of this article, “apparent energy intake” and “apparent macronutrient (carbohydrate, protein, and fat) intake” should be read as “energy available for consumption” and “macronutrient (carbohydrate, protein, and fat) available for consumption”, respectively [15,21].

### 2.3. Macronutrient Distribution Ranges

The macronutrient distribution ranges represent the percentage of energy obtained from carbohydrate, protein, and fat. The Atwater coefficients (4 kcal/g for carbohydrate, 4 kcal/g for protein, and 9 kcal/g for fat) were used to calculate the available energy from the macronutrients [22]. The percentage of the energy contribution from each of the macronutrients was obtained by dividing the energy from a macronutrient by the total available energy. Moreover, the recommended macronutrient distribution pattern in the diet in Bangladesh was derived from the desirable dietary pattern (DDP) for the Bangladeshi population [23]. On the basis of this recommendation, about 55–75% of dietary energy should come from carbohydrates, 10–15% from protein, and 15–30% from fat.

### 2.4. Trends Analysis

To analyze the temporal trends and to colligate significant changes in apparent energy and macronutrient intakes in the diet in Bangladesh, we performed a joinpoint regression analysis by using the Windows-based statistical software, the Joinpoint Regression Program (version 4.8.0.0, National Institute of Health, Bethesda, MD, United States) [24]. With this analysis, it was achievable to identify years when a significant change in the linear slope of the trends in energy and macronutrient intakes was detected over the study period, from 1961 to 2017. The best-fitting points, called “joinpoints”, were selected when the rate changes significantly (*p* < 0.05). In each joinpoint, the trend significantly alters its direction and observes the transformation or changes in energy and macronutrient intakes in the diet. Hence, each joinpoint acts as point of changes in energy and macronutrient intakes and is the basis of the transformation observed. We conceptualized that changes in food availability, technological advancement and policy-driven growth in agriculture, and population growth might have acted as drivers [7] for these joinpoints, the node of changes in apparent energy and macronutrient intakes in the diet.

The analysis began with the minimum number of joinpoints and tested whether one or more joinpoints (in this study up to five) were statistically significant and must be added to the model. To set forth the linear trends by period, we then computed the estimated annual percent change (APC) for each of those trends in the apparent energy and macronutrient intakes in the diet. Moreover, the average annual percentage change (AAPC), calculated as a geometric weighted average of APCs of various segments [25], was used to quantify the trends of apparent energy, macronutrient, and food intake changes in the Bangladeshi diet over the entire period of the available FAOSTAT data (1961–2017). We extended our analysis to an advanced level of the joinpoint regression model, the Jump model, to estimate the effect of the introduction of the news FBS on the underlying trend in apparent energy and macronutrient intakes in the Bangladeshi diet from 1961 to 2017. The Jump model considers the introduction of the new FBS since 2014 that caused a jump in the values, but it is assumed not to affect the underlying trend in apparent energy and macronutrient intakes in the diet in Bangladesh [26].

## 3. Results

### 3.1. Energy

Apparent energy intake (kcal/day/person) in the Bangladeshi diet increased by 1.21-fold with a slower average annual change of 0.3% per year from 1961 to 2017 (Table 1). During these 57 years, apparent energy intake showed a boat-shaped changing pattern (Figure 1A). The apparent intake decreased significantly (*p* = 0.002) by 0.75% per year from 1961 to the early 1970s (Table 2) and reached to the historically lowest point in 1972 (1877 kcal/day/person). After this downward trend, the following intake trend was almost stable with some increase until the late 1990s (APC = 0.15; from 1977 kcal/day/person in 1973 to 2096 kcal/day/person in 1997). Following 1997, within a very short period, apparent energy intake significantly (*p* = 0.033) soared up at a rapid rate until the early 2000s (from 2096 kcal/day/person in 1997 to 2331 kcal/day/person in 2001), with an annual increase rate of 3.12%, and then slowed down and began increasing at a markedly reduced rate (APC = 0.46; *p* < 0.001) until 2017; during these 17 years, apparent energy intake increased by only 11.4%. The proportion of available energy from carbohydrates in the Bangladeshi diet decreased by only 6.8% during the last 57 years (from 84.7% per day per person in 1961 to 78.9% per day per person in 2017), while available energy from fat increased considerably by 72.3% (Table 1).

The carbohydrate energy showed a nearly unchanged (APC = −0.05) and a very little slanting trend from 1961 (84.7% per day per person) to 1980 (84.6% per day per person), and thereafter significantly decreased (APC = −0.18; from 84.5% per day per person in 1981 to 78.9% per day per person in 2017), but the decreasing rate was marginal (Table 2 and Figure 1B). The apparent proportion of energy from protein showed a W-shaped temporal pattern with a very long spread at the outer edge (Figure 1C). The protein energy changed by merely 10.2% per year (AAPC = 0.2) over the last 57 years (1961–2017) and values were within a very narrow range (from 8.4% per day per person to 9.3% per day per person) in the Bangladeshi diet.

During the 1960s, apparent protein energy was almost stable in the diet (APC = −0.18; p = 0.380), and then surprisingly there was a significant (*p* = 0.006) and sharp increase by 1.42% per year until 1973. Since 1973, consecutively for more than two decades, apparent protein energy showed a significant (*p* < 0.001) downward trend by 0.23% per year until 1997 and reached the level prevailing in the 1960s. It was since the late 1990s that the apparent protein energy in the Bangladeshi diet started to increase significantly (*p* < 0.001) by 0.45% per year (from 8.5% per day per person in 1997 to 9.3% per day per person in 2017). The apparent fat energy was virtually unchanged from 1961 (only 6.8% per day per person) to 1979 (only 6.7% per day per person) in the Bangladeshi diet, and unlike carbohydrate energy then increased significantly (*p* < 0.001) at a marked rate with an annual change of 2.27% until the late 1990s (12.9% per day per person in 1998), reaching almost double the level prevailing in the 1960s and 1970s. After 1998, the increasing rate of apparent fat energy in the diet slowed down radically with an annual change of 0.58% until 2017 (11.7% per day per person).

### 3.2. Carbohydrate, Protein, and Fat

The temporal changes in apparent carbohydrate intakes in the Bangladeshi diet from 1961 to 2017 showed a boat-shaped changing pattern (Figure 2A) with similar dynamics and pattern to energy (Figure 1A). For this 57-year time span, the apparent carbohydrate intake did not change significantly (*p* = 0.60) in the Bangladeshi diet and increased by only 0.2% per year on average (Table 2). During the 1960s and early 1970s, apparent carbohydrate intake in the diet showed a downhill trend, decreasing by 0.73% per year (from 456.2 g/day/person in 1961 to 394.6 g/day/person in 1972). Since the early 1970s, the apparent intake slope was almost flat due to the high variability in intake values for more than two and a half decades (APC = 0.02). From 1997, there was a sudden steep increase in apparent carbohydrate intake trend until the early 2000s (APC = 2.38), and since then the rate of increasing trend considerably slowed down (APC = 0.47).

The temporal changes in apparent protein intake in the Bangladeshi diet showed a simple trend with only one joinpoint, appearing like the flight take-off pattern (Figure 2B). The apparent protein intake in the diet increased significantly (AAPC = 0.5; *p* < 0.05) from 1961 (45.5 g/day/person) to 2017 (60.3 g/day/person), and plant sources contributed more than 80% of the protein intake (Table 1). During the 1960s, 1970s, 1980s, and the early 1990s, the apparent protein intake trend in the diet was almost stable with an insignificant (*p* = 0.812) annual percent change (APC = −0.01; from 45.5 g/day/person in 1961 to 43.5 g/day/person in 1994). This long stable protein intake trend was mostly determined by the protein from the plant sources. During this period, about 88% of the apparent protein intake came from plant sources. Moreover, plant protein intake showed a similar steady trend from 1961 (40.1 g/day/person) to 1995 (38.5 g/day/person) characterized by an insignificant (*p* = 0.197) annual change by only 0.08% during this period. During the 1960s, animal protein intake showed an upward trend (APC = 1.23; from 5.45 g/day/person in 1960 to 6.21 g/day/person in 1970) followed by a significant (*p* < 0.001) downward trend by 1.30% per year from 1970 (only 6.21 g/day/person) to 1990 (only 4.98 g/day/person). Since the early 1990s, the apparent protein intake in the Bangladeshi diet increased significantly (*p* < 0.001) by 1.33% per year (from 43.5 g/day/person in 1994 to 60.3 g/day/person in 2017). Both animal and plant protein intakes showed a significant (*p* < 0.001) and upward trend during this increasing apparent protein intake period (Table 2), and the rate of increase was comparatively much higher for animal protein (APC = 3.43) than plant protein (APC = 0.94).

The changing trend of apparent fat intake is almost similar to the trend of apparent protein intake in the diet (Figure 2C). The average growth rate of apparent fat intake in the Bangladeshi diet from 1961 to 2017 was 1.4% per annum (95% CI: 1.1 to 1.7) and it was increased by 2.08 times during these 57 years. During the 1960s and 1970s, apparent fat intake in the Bangladeshi diet showed an insignificant (*p* = 0.085) downward trend (APC = −0.59; from 16.3 g/day/person in 1961 to 14.0 g/day/person in 1977). This was the case since the late 1970s, when apparent fat intake in the diet started to increase significantly (*p* < 0.001) at a marked rate. During this period, apparent fat intake in the diet increased considerably by 2.16% per year (from 13.6 g/day/person in 1977 to 33.9 g/day/person in 2017). Fat from plant sources mostly dictated this trend and increased by almost 10.0% (from 70.2% in 1977 to 80.3% in 2017) during this period. Similar to apparent fat intake, plant fat intake significantly (*p* < 0.001) increased by 3.20% between the late 1970s and the late 1990s and since then continued to increase at a slower rate (APC = 1.29; from 23.1 g/day/person in 2000 to 27.3 g/day/person in 2017). Fat intake from animal sources increased by only 0.23% per year during the 1960s (average intake was only 3.9 g/day/person) and 1970s (average intake was only 4.0 g/day/person), followed by a sharp downward trend (APC = −6.88) from 1979 (4.27 g/day/person) to 1982 (3.46 g/day/person). Since then, apparent intake of fat from animal sources showed an upward trend from 1982 to 2000 (APC = 1.40), followed by a marked (*p* < 0.001) steep increase by 2.88% per year up until 2017 (6.68 g/day/person).

## 4. Discussion

In this study, we analyzed the temporal trends and characterized the significant changes that have taken place in the apparent energy and macronutrient consumption in the diet in Bangladesh from 1961 to 2017. For the very first time, we analyzed the apparent energy and macronutrient intake trends in the Bangladeshi diet for the 57-year time span by combining the FAO’s old and new food balance sheets (FBS) data. To our knowledge, this is the first study ever that used the new FBS data after it was first published on December 19, 2019 [17]. It reports on the dynamics and temporal trends in apparent energy and macronutrient intake in Bangladesh in an omnibus way from 1961 to 2017 by using joinpoint regression analysis considering and adjusting the introduction of new FBS data since 2014. In our analysis, we found that the introduction of new FBS causes an increase or jump in apparent energy and carbohydrate intake values by only 3% (comparability ratio, CR = 1.03), whereas for apparent protein (CR = 1.01) and fat (CR = 0.98) intakes, the effects are almost same and in marginal quantity. Hence, trends in apparent energy and carbohydrate intakes were analyzed on the basis of the standard joinpoint model and jump model where the effect of the introduction of the new FBS since 2014 was considered and assumed to not affect the underlying trend in apparent energy and macronutrient intakes in the diet in Bangladesh. Above all, this is the first and the longest historical energy and macronutrient availability data analysis of Bangladesh.

The population of the eighth-most populous country of the world, Bangladesh, has been suffering from chronic energy deficiency during the 1960s, 1970s, 1980s, and 1990s, and apparent energy intake was less than 2200 kcal/day/person. Since the late 1990s, Bangladesh has made remarkable progress in raising the apparent energy consumption in the diet. Eventually, apparent energy intake increased by 20.5% in Bangladesh over the last 57 years (1961–2017), reaching almost 2600 kcal/day/person in 2017. During the 57-year time span, apparent energy intake showed a boat-shaped changing pattern that was solely determined and almost similar to the changing trends of carbohydrates; however, carbohydrates continued to remain by far the most over eaten macronutrient in the Bangladeshi diet. Even in 2017, carbohydrates contributed to about 79% of the apparent energy intake in the diet. Apparent intake trends of energy and macronutrients were disrupted (energy and carbohydrate) and stable with an inadequately low base (protein and fat) before, during, and following the Liberation War of 1971. Following 1980, a decreasing trend of carbohydrate energy and an increasing trend of fat energy were observed in the Bangladeshi diet. Since the late 1970s, apparent fat intake in the diet started to increase significantly at a marked rate by 2.16% per year, whereas since the early 1990s, the apparent protein intake increased significantly by 1.33% per year. Plant sources mostly dictated the protein and fat intake trends in the Bangladeshi diet since 1960, whereas animal sources started to contribute significantly since 1990 in protein intake by 3.43% per year and since 2000 in fat intake by 2.88% per year.

### 4.1. Trends in the Apparent Intake of Energy

Apparent dietary intake, in terms of per capita daily energy (kcal) intake, is the key variable used for measuring and assessing the evaluation of the worldwide food situation [15]. Moreover, apparent energy intakes can describe the nutrition transition at the most general level [2]. Our analysis revealed that Bangladesh has experienced a late energy revolution in their dietary history. During the 1960s, 1970s, 1980s, and 1990s, Bangladesh was suffering from substantive calorie deficits, and apparent energy intake was less than 2200 kcal/day/person. Since the late 1990s, Bangladesh gained remarkable momentum in raising the apparent energy consumption in the diet, which slowed down later. At the beginning of the 1960s, the entire developing world, apart from the only a handful of countries, was suffering from substantial calorie deficits, chronic malnutrition, and population-wide famine. During this period, Asia in particular was frequently hit by famine, affecting large segments of its population [2]. In the case of Bangladesh, during the 1960s and 1970s, food availability was seriously disrupted before, during, and immediately after the Liberation War of 1971, as well as following natural disasters such as cyclones, droughts, and floods [7]. The population of Bangladesh has been known to be in a state of chronic malnutrition for many decades, and the enormity of this problem was first documented in a benchmark nutrition survey carried out during 1962–1964 [28]. That benchmark survey quantified the prevalence and effects of widespread malnutrition on the population of Bangladesh during the 1960s, evidenced by physical signs and biochemical markers of deficiency and gaps between intake and requirements. Protein energy malnutrition, vitamin A deficiency, riboflavin deficiency, anemia, goiter, and vitamin C deficiency were reported as specific problems of the population [11]. The changes in apparent energy intake have influenced the nutrition and health of the population in Bangladesh. In our analysis, the first trend in apparent energy intake spanned from 1960 to just after the liberation war of Bangladesh (1971), being characterized by a decline in the availability of dietary energy. During this period, there had been a detectable decline in per capita food production, with a distinct downward trend in per capita availability of food [29]. Moreover, domestic food production in Bangladesh has failed to keep up with food demand generated by rapid population growth over this period. As a result, there has been a perceptible decline in per capita food availability and consumption [7]. The nutrition and health implications of this change were well documented in the benchmark survey during 1962–1964. The survey reported that about 46% of the studied households had inadequate energy intake, which was supported by the skin fold thickness measurements of the population that indicated that the population was thin and had little subcutaneous fat [11]. In addition, this declining trend in apparent energy intake also affected the child nutrition outcomes during the 1960s. The growth curve of Bangladeshi children aged 0–4 and 5–18 years was well below the 16th percentiles of reference growth curve of children of European descent, and during the growth spurts, the two curves deviated further, which indicated a further deviation of weight and height of Bangladeshi children from the reference children of European descent [11]. On the other hand, during this period, DR-NCDs were not a major public health problem in Bangladesh; the overall prevalence of diabetes was only 1.5% among the population. Moreover, only 2.7% of males and 1.1% of females over 15 years of age had glucoseuria. Furthermore, 5.3% of the studied population (28 people) had cholesterol levels of 200 mg/100 mL of blood or greater [11].

In the early 1970s, Bangladesh continued to be considered as a food-deficit country with a population of about 75 million. During 1970–1974, a noticeable decline in per capita calorie consumption was noted (about 6% from 1960–1965) [29] due to Bangladesh’s War of Independence and its aftermath. An analysis of the pre-war (1970) and post-war (1973) direct cost estimation showed that Bangladesh incurred USD 9.53 billion in direct cost and USD 14.08 billion indirect cost of this war [30]. Destruction of physical capital and human capital loss due to the brutality of Pakistani armies were considered in the direct cost measurement. During the post-liberation war period (1972), the per capita availability of daily energy was the lowest in the dietary history of Bangladesh (only 1877 kcal/day/capita). In the post-liberation period, during the 1970s and 1980s, apparent energy intake was inadequate in the diet. Since the liberation war until the late 1990s, the available energy increased at a marginal rate by 0.17% per year. The Bangladesh National Nutrition Surveys (in 1975–1976 and 1981–1982) alerted to the growing deterioration of nutritional status, which was accompanied by less and less energy intake in the diet of the Bangladeshi population. Energy intake was 2094 kcal/day/person in 1975–1976, which reduced to 1943 kcal/day/person in 1981–1982 [10]. Widespread prevalence of protein energy deficiency in rural Bangladesh was also reported in a secondary data analysis of the National Nutrition Survey of Rural Bangladesh in 1975–1976 [31]. The total prevalence of energy deficiency was around 59% and almost half of the energy-deficient households also had protein deficiency in the diet [31]. This energy deficiency at the population level increased the prevalence of malnutrition, which was confirmed from the increased level of retarded physical growth of the children during the 1970s and early 1980s. On the basis of Gomez classification, in 1975–1976, about 26% of the under-five children were found severely malnourished, which was reduced to 15% in 1981–1982. Alarmingly, the prevalence of moderate malnutrition in under-five children was 53% in 1975–1976 and around 46% in 1981–1982 [28]. Malnutrition was more prevalent among the children aged 5–11 years. About 36% of the 5–11-year-old children were found severely malnourished in 1975–1976, which reduced to 29.4% in 1981–1982 [28].

Since the late 1990s, there was a significant increase in energy availability in the diet of Bangladeshi people. From the second half of the 1990s up to the first half the 2000s, the per capita daily energy availability in the diet increased annually by 60 kcal. During this period, energy availability increased from 2003 kcal/day/capita (in 1994) to 2382 kcal/day/capita (in 2004). This increasing trend of available energy in the 1990s in the Bangladeshi diet can be explained by the advancement and diffusion of new agriculture technology introduced since the second half of the 1980s [7,32]. Moreover, unlike most other Asian countries, Bangladesh had a sluggish start in terms of the Green Revolution and in the 1990s saw dramatic changes in the government policy [33], which eventually caused a remarkable increase in apparent energy intake in Bangladesh since the late 1970s. Since 1990, the government progressively liberalized the agriculture inputs; specifically, small-scale irrigation equipment import policy. This resulted in an outstanding water control afforded by the irrigated *boro* crops, wherein there was an accelerated yield in *boro* crops [7,33]. Since the mid-1960s, the per capita available energy has been steadily increasing on a worldwide basis; available energy from the mid-1960s to the late 1990s increased globally by 450 kcal/day/capita and by over 600 kcal/day/capita in developing countries [34]. Furthermore, from 1997 to 2015, the per capita food consumption (kcal/day/person) in South Asia increased from 2403 kcal/day/person to 2700 kcal/day/person [35]. After the year 2000, the per capita daily available energy moved to a slower increasing phase with an increase of 10.0 kcal/year from 2001 to 2013. A re-analysis of the Household (Income and) Expenditure Survey (H(I)ES 2010) data showed that the per capita actual mean energy intake of the Bangladeshi population was 2190 kcal/day [23]. Moreover, analyzing the same dataset, the Bangladesh Bureau of Statistics (BBS) reported that the per capita actual mean energy intake of the Bangladeshi population was 2308 kcal/day [27]. The recent and the latest Household (Income and) Expenditure Survey (H(I)ES 2016) report by the BBS showed that per capita per day energy consumption decreased to 2210 kcal in 2016 and decreased by 4.23% from 2010 [27]. This reduced actual energy consumption was due to the reduction of per capita daily food intake by 2.45% at the national level from 2010. The remarkable progress in apparent energy intake since the late 1990s was mainly achieved due to the high rice yields, which is the product of late Green Revolution in Bangladesh. These rice yields have ample and statistically meaningful associations with child weight gain [33]. During 1997–2011, the 0.5 metric ton/acre increase in yields predicted a 0.4 SD improvement in the weight-for-height Z score. Moreover, the same increase in yields predicted a 12% decline in mild wasting [33]. The increasing trend in apparent energy intake might have influenced the nutritional status of children and women in Bangladesh. The prevalence of stunting, underweight, and wasting decreased by around 19% (from 60% to 41.2%), 16% (from 52.2% to 36.2%), and 5% (from 20.6% to 15.5%) between 1996 and 2011, respectively [36]. In 1997, 52% of women had chronic energy deficiency, which dropped to 45% in 2000, 38% in 2004, 30% in 2007, and 24% in 2011 [37]. Bangladesh has experienced a considerable social and economic development. The per capita monthly income increased by 208% from 1995–1996 to 2010 and percentage of poor decreased by around 25% between 1991 and 2010. These might have also contributed to the improvement in maternal and child nutritional status in Bangladesh [38].

### 4.2. Trends in the Apparent Carbohydrate, Protein, and Fat Intakes

Carbohydrates have been the most important and the bulkiest macronutrient for providing energy in the diet. In the developing countries, staples are the major contributor to carbohydrates in the diet. In case of Bangladesh, a re-analysis of the H(I)ES 2010 data reported that around 40% of the population had more than 75% of their total energy intake from carbohydrate alone [23]. In line with the re-analysis of this national dataset, our analysis showed that carbohydrate alone had been contributing almost over 78% of the apparent energy share in the diet in Bangladesh during the 57-year time span, since 1961. The structural similarity of energy and carbohydrate intakes revealed that the apparent energy intake was largely determined by the apparent carbohydrate consumption in the diet. This similarity also revealed the lack of sufficient diversity in the diet with low availability of protein and fat over the 57-year period. Cereals, exclusively rice, have been the chief source of carbohydrates in the Bangladeshi diet, like other developing countries, particularly in Asia and parts of Africa [39]. The likeness of the apparent cereal and carbohydrate intake trend in the Bangladeshi diet revealed the commanding role of cereals in the diet and mostly influenced and explained the trends in carbohydrate availability in the diet [7].

From the beginning of the 1960s, the agro-industrial revolution in the developed world in the previous century got a foothold in the food and agriculture sectors of the most advanced developing countries [32]. However, Bangladesh did not experience this agro-industrial revolution until the 1980s, which was reflected in the availability of energy and macronutrients in the diet. A particularly significant pattern that started to rise up in the 1980s was the change in the amount of dietary carbohydrate and fat energy supplies. A decreasing pattern of carbohydrate energy and increasing pattern of fat energy were observed after 1980. The present results show that during the early 1960s the food supply had a very high carbohydrate share and was very low in fat. More than 80% of the total energy (kcal/day/person) came from carbohydrate and less than 10% of the daily energy was from fat. By the early 1980s, the gap between energy from carbohydrate and energy from fat had started to reduce. This period can be treated as a transition period of carbohydrate and fat energy in the available diet of the Bangladeshi population. The re-analysis of the 2010 H(I)ES data showed that around 40% of the population had actual carbohydrate intake more than the recommended energy intake from carbohydrates (75%) and about 21% of the Bangladeshi population had carbohydrate intake more than 400 g/day [23]. Moreover, 40% of the total population consumed less than 10% of the total energy from protein and 53% of the population took less than 15% of total energy from fat [23]. Both the apparent and actual intake of macronutrients in the diet showed imbalance and deviation from the recommended intake level. The recommendation for Bangladeshi people is that about 55–75% of dietary energy should come from carbohydrates and 15–30% should come from fat [23]. Furthermore, in 2002, the Institute of Medicine developed acceptable macronutrient distribution ranges (AMDRs) for carbohydrates, protein, and fat, setting forth an evidence-based range [40]. These guidelines were designed to ensure an adequate intake of nutrients and to address the relation of macronutrients and DR-NCDs. The balance intake of macronutrients within the acceptable ranges has a role in reducing the DR-NCDs [41,42]. The intake values in the diets in the Bangladeshi population are at the far end of the acceptable macronutrient distribution ranges (AMDRs) for carbohydrates, protein, and fat. The shift towards increased energy supply from carbohydrates and low energy supply from protein and fat revealed the imbalance and deviation from AMDRs in the available diet in Bangladesh. Bangladesh experienced a large increase in overweight, and the DR-NCDs now impose a major health burden in Bangladesh; in terms of lives lost due to ill-health, disability, and premature death, DR-NCDs accounts for 61% of the total disease burden [6]. In addition, the prevalence of overweight among Bangladeshi women increased from 5% in 2000 to 17% in 2011 [37]. The mean BMI increased from 20.2 in 2004 to 21.4 in 2011, whereas the proportion of women who were overweight or obese increased to almost double (from 9% in 2004 to 17% in 2011) [37]. In the case of children under five, overweight and obesity incidence increased from 0.7% in 2007 to 1.5% in 2011, signaling that Bangladesh is gradually experiencing a nutritional shift among the children in this age range [38]. Apparent protein intake in the diet increased by about 15 g/day/person in an absolute amount from 1961 to 2017, and plant sources have been contributing more than 80% of the protein intake. During the 1960s, 1970s, 1980s, and the early 1990s, the apparent protein intake trend in the diet was almost stable with only 0.01% changes (from 45.5 g/day/person in 1961 to 43.5 g/day/person in 1994). This long stable protein intake trend was mostly determined by the protein from the plant sources, which accounted for about 88% of the apparent protein intake. In 1960–1965, per capita daily protein intake averaged 43.4 g and was almost 80% from cereals [29]. By 1965–1970, protein intake reduced modestly, and the decline accelerated during the early 1970s [29]. The first nutrition survey in Bangladesh during 1962–1964 reported that 64% of the households had inadequate protein intake. Almost 70% of the dietary protein was cereal protein and only 14% was from animal sources [11]. In a population existing on a deficient diet, multifaceted deficiencies are more typical than deficiency of any single nutrient. For example, keratomalacia is frequently found to be associated with kwashiorkor, and the medical prognosis of protein deficiency is significantly more detrimental in the presence of keratomalacia [11]. Foods of animal origin, especially milk and milk products, are the best sources of riboflavin and iron. Almost half of the population had deficient or low levels of urinary riboflavin excretion [11], suggesting that riboflavin deficiency was widely prevalent among the population during the 1960s. Anemia, mostly iron deficiency anemia, is a general and severe problem in Bangladesh. About 45% of male children and 50% of female children were found to have hemoglobin concentration below 12 mg/100mL of blood. Alarmingly 60% of the pregnant and lactating women had low hemoglobin levels, with 80% of the anemia being caused by iron deficiency [11].

Since the early 1990s, the apparent protein intake in the Bangladeshi diet increased considerably by 1.33% per year. Both animal and plant protein intakes increased significantly since the early 1990s and the rate of increase was comparatively much higher for animal protein (3.43% per year) than plant protein (0.94% per year). This trend in apparent animal protein intake was influenced and mostly can be explained by the availability of fish in Bangladesh [7]. Fish is an important animal food in Bangladesh because of its increased availability, as well as its high micronutrient and protein content. The apparent protein intake trend in the Bangladeshi diet is approaching close to the recommended level, but more than 80% of the apparent protein intake is contributed by plant sources. The quality of the protein is poor, as even in cases of actual intake at the population level, around 75% of protein comes from plant sources [23]. During the 1990s, more than 86% of the available protein was supplied by the plant sources, whereas animal sources contributed only 11–13% of the protein intake. Moreover, since 2015, protein contribution from animal sources increased to 20% of the apparent intake. Cereals, largely rice availability at the national level, influenced the apparent plant protein intake trend, and fish availability at the national level influenced the animal protein intake trend in the population diet in Bangladesh [7]. The continued and rapid growth of aquaculture over the past 30 years has resulted in a 4.1-fold increase in total fish production in Bangladesh [43]. Moreover, since 1990, there has been a considerable increase in milk, meat, and egg production in Bangladesh [32]. The increasing consumption of animal food since 1990 might have acted to reduce the protein energy malnutrition in children under five in Bangladesh. The prevalence of underweight in 1989–1990 was more than 65% and came down to 47% in 2000 and around 36% in 2011.

The rise in the amount and nature of the fats consumed in the diet is a significant component of nutrition transition, with there being huge variations across areas of the world in terms of fat intake levels [44]. The average global supply of fat increased by 20 g/day/person since 1967 until the last of the previous century, and this increase was most pronounced in the Americas, the European community, and East Asia [45]. The largest share of dietary fat in the diets of developing countries comes from vegetable oils, which have shown a very high consumption growth in developing countries [1]. Like other developing countries, Bangladesh experienced a 2.6-fold increase in the apparent consumption of vegetable oils from 1961 to 2013. This increased amount of apparent vegetable oil intake determined the fat availability trend in the diet [7]. Fat from plant sources, mainly vegetable oils [7], mostly dictated the fat intake trend, and since the late 1970s has increased by almost 10% (from 70.2% in 1977 to 80.3% in 2017). Fat intake from animal sources increased by only 0.23% per year during the 1960s and 1970s, and during this period, on average about 26% of the fat was sourced from animal foods. Since the early 1980s (average intake of only 3.50 g/day/person), apparent intake of fat from animal sources showed an upward trend, followed by a marked steep increase by 2.88% per year up until 2017 (only 6.68 g/day/person).

The rising consumption in per capita intake in Bangladesh is not always a sheer blessing. The related diet transition usually applies changes towards high energy-dense fat diets, particularly those high in saturated fat, sugar, and salt, and low in unrefined carbohydrates. In combination with unbalanced macronutrient intake and unbalanced proportion of macronutrients in the diet with lifestyle changes, rapid urbanization is often accompanied by a corresponding increase in DR-NCDs. The problem of DR-NCDs and acute and chronic malnutrition co-exist, and these countries are confronted with a double burden of malnutrition. A report from the Global Burden of Disease (GBD) shows that malnutrition was the number one health risk factor and cause of mortality in children aged under five years old in Bangladesh in the 1990s, whereas around 51% reduction of the rate of this health issue was observed in 2017 [46]. Similarly, prior to the year 2000, which corresponds to the period of calorie deficit in the Bangladeshi diet, unlike DR-NCDs, infectious and diarrheal diseases have been among the major causes of death. However, since 2002–2004, an epidemiological transition has been observed in Bangladesh, with an increase in rates of DR-NCDs (diabetes, cardiovascular diseases) and decrease in the prevalence of diseases that are associated with malnutrition such as infectious diseases (−44.9% for respiratory infectious diseases and −6.5% for diarrheal diseases in 2017) [46]. In fact, malnutrition causes immunodeficiency and reduces the capacity of the human organism to fight against diseases, infections in particular [47]. Thus, the decrease in under-five mortality as well as that of rates of infectious diseases could, at least partially, be explained by the improvement of nutritional status (increase in energy intake) in Bangladesh in the last decades. 

We have summarized the significant changes in apparent energy and macronutrient intakes in the Bangladeshi diet in Figure 3. Bangladesh has experienced a late energy revolution in its dietary history. From the 1960s to the 1990s, Bangladesh was suffering from dietary energy deficiency as per capita energy availability was less than 2200 kcal/day. Since the late 1990s, Bangladesh has made remarkable progress in raising the apparent energy consumption in the diet. Energy availability increased by 20.5% in the diet from 1961 to 2017. Carbohydrate energy availability decreased by only 6.8%, while available fat energy increased by as much as 72.3% in the diet. Moreover, carbohydrate alone contributed almost more than 78% of the energy share in the diet during this 57-year time span. The carbohydrate energy showed nearly stable downward trends from 1961 to 1980 and, thereafter, significantly decreased at a very slow rate. Since the late 1970s, apparent fat intake in the diet started to increase significantly at a marked rate by 2.16% per annum, whereas from the early 1990s, the apparent protein intake increased significantly by 1.33% per year.

The FAO’s food balance sheet data have some shortcomings that need to be addressed. Energy and macronutrient intake data derived from this source do not indicate the actual consumption of energy, carbohydrate, protein, and fat. This is an average quantity of energy and macronutrients at the national level that was available for consumption at the household or individual level. Therefore, it is unfeasible from our study to make an inference on individual or sub-national level energy, carbohydrate, protein, and fat intake trends and to study the inequalities. Different confinements of this source are some practical issues, for example, coverage and representativeness of the basic data since most of the statistics developed are restricted to commercialized major food crops. Non-commercial or subsistence-level production, normally frequent in poor areas, was excluded. There is a potential issue of overestimating energy, protein, and fat consumption since the FBS does not take into account food losses that happen after the retail level. Food that is spoiled while processing at the household level, e.g., wasted trimmings, and as is regular practice in the rural parts of Bangladesh, food given to domestic animals inside the households, were not accounted for in the computation of the FBS. Another limitation is that there was a large disadvantage in enumerating the fruit and vegetable categories in the FBS, which might affect the energy calculation to some degree. In addition, in the old food balance sheet (reported from 1961 to 2013) inherited most of its statistical error related to the stocks and industrial utilization or feed values, which resulted in the outstanding unbalanced amounts in the FBS. Since 2014, in the new FBS, the advancement in computational quality, with a balancer variable and imputations, has reduced most of the statistical errors in the old FBS. Despite these limitations, the FBS is the only cost-effective tool for analyzing the temporal trend and for long-running comparisons of dietary changes at a national level in Bangladesh. Studies indicate that food balance sheets are useful for temporal trend analysis as the databases are standardized and updated regularly [48]. Moreover, the use of FBS data to analyze the trends in nutrient availability in a country is more reliable than using absolute values at a single point in time [48]. Food balance sheets are still a very good available data source for the analysis of energy and macronutrient intake changes during a particular period of time for a given country. Furthermore, with a wider view, FBS data, together with dietary intake data and with a wide range of other facts, can describe the nutrition and disease transition and their connection with agricultural revolution and economic growth [49].

## 5. Conclusions

In this study, we revealed the temporal trends in apparent energy and macronutrient intake in the diet in Bangladesh from 1961 to 2017, the largest historical analysis of the apparent intake data in Bangladesh to date. We found that since the late 1990s, Bangladesh has made significant progress in raising the apparent energy consumption in the diet and eventually energy adequacy in the diet was improved. Carbohydrate alone contributed to almost over 78% of the energy share in the diet. With the bulk amount of carbohydrate intake, apparent protein and fat intake levels have also been increasing in the diet. Since the late 1970s, apparent fat intake in the diet started to increase significantly at a marked rate by 2.16% per annum, whereas since the early 1990s, the apparent protein intake increased significantly by 1.33% per year. This increasing apparent consumption of protein and fat gave signs of structural changes in the diet with the substitution effect and dietary diversity involving increasing availability of fish, eggs, meat, milk, and vegetable oil. These structural changes have increased the dietary diversity, but the amount was grossly inadequate to have any positive effect on health. Most of these changes were related to the expansion effect and were characterized by higher energy supply from low-cost plant foods, mostly from cereals. The substitution effect, where shifts from carbohydrate-rich staples to increased protein and fat consumption occur, began in the diet, but the amount was grossly inadequate, even at the national level. Summed up briefly, Bangladesh has defeated the substantive calorie deficit condition in the diet since the late 1990s. However, grossly inadequate structural changes such as excessive carbohydrates along with imbalanced and low-quality protein and fat intakes have been the features in the diet in Bangladesh. This inadequate and imbalanced dietary intake may act as the prime cause behind the increasing prevalence of overweight or obesity and the emergence of DR-NCDs in Bangladesh.

## Figures and Tables

**Figure 1 nutrients-12-02319-f001:**
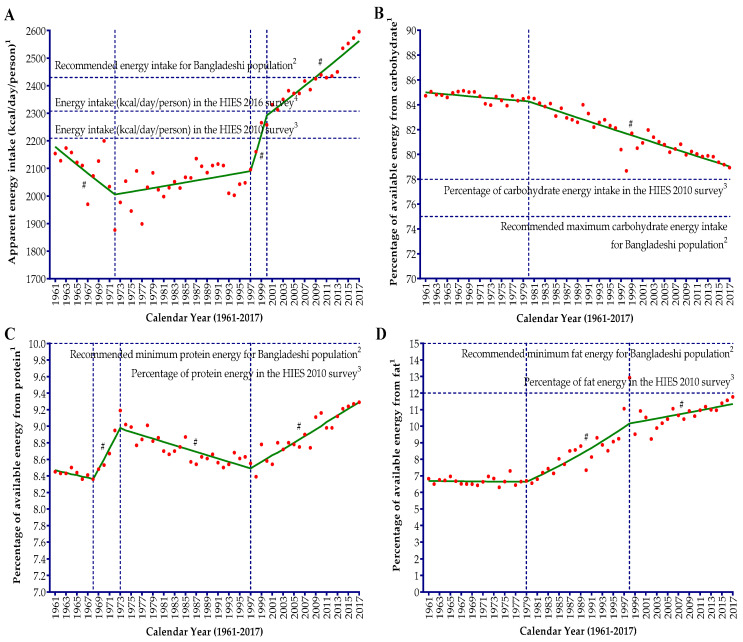
Joinpoint regression analysis of the apparent intake of energy (**A**) and the percentage of available energy from carbohydrates (**B**), protein (**C**), and fat (**D**) in the diets of the Bangladeshi population from 1961 to 2017. A vertical dotted line represents the joinpoints. ^1^ Apparent energy intake refers to the average availability of energy in the diets for consumption and percentage of available energy from carbohydrates (**B**), protein (**C**), and fat (**D**). ^2^ A horizontal dotted line represents the recommended energy intake and recommended percent distribution of available energy from macronutrients on the basis of the desirable dietary pattern (DDP) for the Bangladeshi population [23]. ^3^ A horizontal dotted line represents the mean actual energy intake (kcal/day/person) and macronutrient distribution to energy (carbohydrates (**B**), protein (**C**), and fat (**D**)) in the diet of the Bangladeshi population on the basis of the dietary analysis of the Household (Income and) Expenditure Survey (H(I)ES) in 2010 [23]. ^4^ A horizontal dotted line represents the mean per capita actual energy intake (kcal/day) in the diet of the Bangladeshi population on the basis of the dietary analysis by the Bangladesh Bureau of Statistics (BBS) of the latest H(I)ES in 2016 [27]. # denotes the annual percent change (APC) that is significantly different from 0 (*p* < 0.05).

**Figure 2 nutrients-12-02319-f002:**
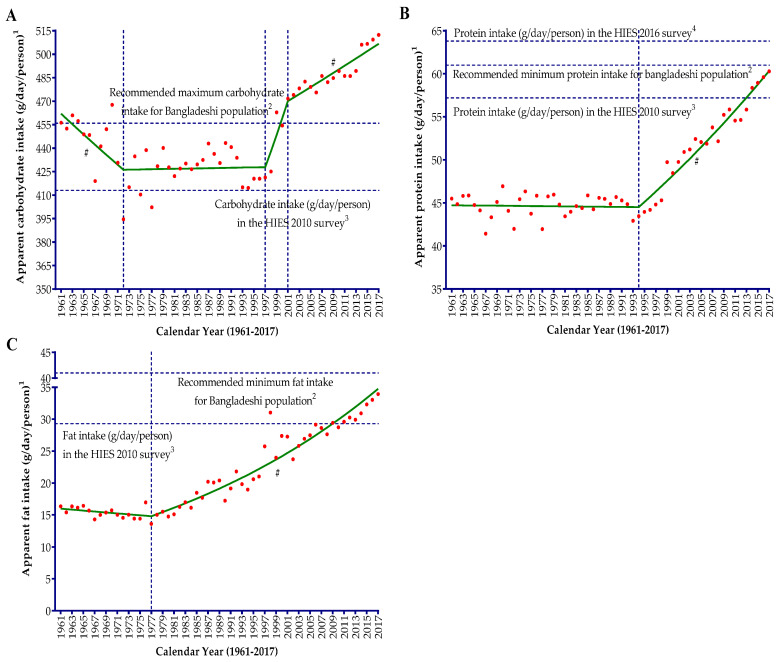
Joinpoint regression analysis of the apparent intakes of carbohydrate (**A**), protein (**B**), and fat (**C**) in the diets of the Bangladeshi population from 1961 to 2017. A vertical dotted line represents the joinpoints. ^1^ Apparent intakes of carbohydrate (**A**), protein (**B**), and fat (**C**) refer to the availability of carbohydrate (**A**), protein (**B**), and fat (**C**) in the diets for consumption in the Bangladeshi population. ^2^ A horizontal dotted line represents the recommended carbohydrate (**A**), protein (**B**), and fat (**C**) intakes on the basis of the desirable dietary pattern (DDP) for the Bangladeshi population [23]. ^3^ A horizontal dotted line represents the mean actual carbohydrate (**A**), protein (**B**), and fat (**C**) intake (g/day/person) in the diet on the basis of the H(I)ES in 2010 [23]. ^4^ A horizontal dotted line represents the mean actual protein intake (g/day/person) in the diet of the Bangladeshi population on the basis of the dietary analysis by the Bangladesh Bureau of Statistics (BBS) of the latest H(I)ES in 2016 [27]. # denotes the annual percent change (APC) that is significantly different from 0 (*p* < 0.05).

**Figure 3 nutrients-12-02319-f003:**
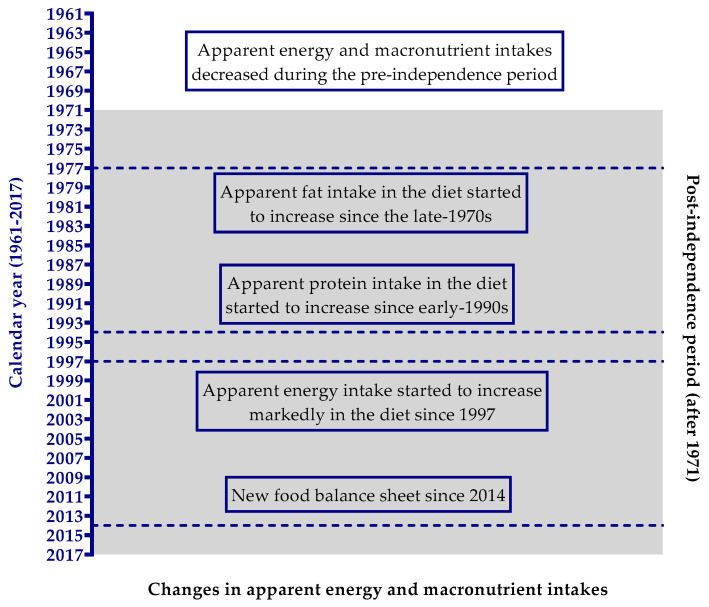
A brief summary of the significant changes over the periods in apparent energy and macronutrient intakes in Bangladesh from 1961 to 2017. Shaded area indicates the time after independence and the unshaded area represents the time before independence. The horizontal lines represent the year where significant changes started.

**Table 1 nutrients-12-02319-t001:** Apparent consumption of energy and macronutrient in Bangladesh, 1961–2017.

Energy and Macronutrinet	1961	2017	% Change ^1^	Status ^2^	*p*-Value ^3^
Energy (kcal/day/person)	2154	2596	20.52	Increased	<0.05
Macronutrient distribution					
Carbohydrate (%)	84.73	78.94	6.83	Decreased	<0.05
Protein (%)	8.45	9.29	9.94	Increased	<0.05
Fat (%)	6.83	11.77	72.33	Increased	<0.05
Macronutrient (g/day/person)					
Carbohydrate	456.25	512.33	12.29	Increased	0.60
Protein	45.49	60.28	32.51	Increased	<0.05
Animal protein	5.45	11.96	119.44	Increased	<0.05
Plant protein	40.04	48.33	20.70	Increased	<0.05
Fat	16.34	33.95	107.77	Increased	<0.05
Animal fat	3.84	6.68	73.96	Increased	<0.05
Plant fat	12.50	27.27	118.16	Increased	<0.05

^1^ Values of the changes in apparent intake level are given as an absolute figure; directions of the changes from 1961 until 2017 are reported in the status. ^2^ Status has two responses: increased (when apparent intake increased from the intake level of 1961) or decreased (when apparent intake decreased from the intake level of 1961); the status is based on the direction of the percent change from 1961 to 2017, where 1961 is considered as a baseline year and 2017 is the latest available year. ^3^
*p*-values are based on the average annual percent change of the different segment of the trend from 1961 to 2017, considering the introduction of new food balance sheets since 2014 by using the Jump model.

**Table 2 nutrients-12-02319-t002:** Trends in apparent energy and macronutrient intakes in the diet in Bangladesh from 1961 to 2017 ^1^.

	AAPC ^3^	Trend 1		Trend 2		Trend 3		Trend 4	
	1961–2013	Period	APC ^2^	Period	APC ^2^	Period	APC ^2^	Period	APC ^2^
Energy									
Joinpoint model ^4^	0.3 (0.0 to 0.7) **	1961–1972	−0.75 *	1972–1997	0.17 *	1997–2000	3.12 *	2000–2017	0.66 *
Jump model ^5^	0.2 (0.0 to 0.5) **	1961–1972	−0.73 *	1972–1996	0.15 *	1996–2001	2.27 *	2001–2017	0.46 *
Macronutrient distribution									
Carbohydrate	−0.1 (−0.2 to −0.1) **	1961–1980	−0.05	1980–2017	−0.17 *				
Protein	0.2 (0.1 to 0.3) **	1961–1968	−0.18	1968–1973	1.42 *	1973–1997	−0.23 *	1997–2017	0.45 *
Fat	0.9 (0.7 to 1.2) **	1961–1979	−0.04	1979–1998	2.27 *	1998–2017	0.58 *		
Macronutrient									
Carbohydrate intake ^4^	0.2 (−0.1 to 0.4)	1961–1972	−0.73 *	1972–1997	0.02	1997–2001	2.38	2001–2017	0.47 *
Carbohydrate intake ^5^	0.1 (−0.3 to 0.5)	1961–1972	−0.75 *	1972–1998	0.03	1998–2001	3.41	2001–2017	0.25
Protein intake	0.5 (0.4 to 0.6) **	1961–1994	−0.01	1994–2017	1.33 *				
Animal protein	1.3 (1.1 to 1.6) **	1961–1970	1.25 *	1970–1990	−1.33 *	1990–2017	3.43 *		
Plant protein	0.4 (0.3 to 0.5) **	1961–1995	0.08	1995–2017	0.94 *				
Fat intake	1.4 (1.1 to 1.7) **	1961–1977	−0.48	1977–2017	2.16 *				
Animal fat	1.0 (0.6 to 1.4) **	1961–1979	0.23 *	1979–1982	−6.88 *	1982–2000	1.40 *	2000–2017	2.88 *
Plant fat	1.4 (1.0 to 1.7) **	1961–1977	−0.84 *	1977–1998	3.20 *	1998–2017	1.29 *		

^1^ Trend analysis identified joinpoints, which are points where line segment of trends are joined. Each joinpoint denotes a statistically significant change (*p* = 0.05) in trend. ^2^ APC is the annual percent change within a trend in energy, macronutrient distributions (MDs), and macronutrient intake in the diet. ^3^ AAPC is the average annual percent change in energy, macronutrient distributions (MDs), and macronutrient intakes in the diet, calculated as a geometric weighted average of the calculated APCs of various segments from 1961 to 2017; in parentheses, the 95% confidence interval is presented. ^4^ Trends were analyzed on the basis of the standard joinpoint model where the introduction of the new food balance sheet since 2014 was not considered; on the basis of our analysis, the introduction of a new food balance sheet caused only a 3% (comparability ratio, CR = 1.03) increase in the values, mainly in energy and carbohydrate intake. ^5^ Trends were analyzed on the basis of the Jump model where the effect of the introduction of the new food balance sheet since 2014 was considered. * Denotes that the annual percent change, APC, was significantly different from 0 for a specific trend (two-sided *p* < 0.05) adjusted with the Jump model due to the introduction of new food balance sheet data from 2014. ** Denotes that the average annual percent change, AAPC, was significantly different from 0 for the entire trend from 1961 to 2017 (two-sided *p* < 0.05) adjusted with the Jump model due to the introduction of new food balance sheet data from 2014.

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
