# Peer review of "Temporal Trends in Apparent Energy and Macronutrient Intakes in the Diet in Bangladesh: A Joinpoint Regression Analysis of the FAO’s Food Balance Sheet Data from 1961 to 2017"

_nutrients, 2020, doi:10.3390/nu12082319_

Round 1
Reviewer 1 Report
The current manuscripts aims to analyze the temporal trends and changes apparent energy and macronutrient consumption in Bangladesh from 1961 to 2017 and to estimate the projected apparent energy and macronutrient intakes in the diet in Bangladesh unto 2050. The manuscript has a clear structure and a method section with appropriate information to allow replication. The analyses of the dynamics and temporal trends in apparent energy and macronutrient intake in Bangladesh is highly pertinent.
My major consideration and the weakness of the current manuscript in the present form is the lack of nutritional or health implications of the findings. The outcome measures (trends in apparent energy and macronutrient intake) are shown in the results, discussed in relation to historical events in Bangladesh for the various subsections and repeated again in the discussion. Especially in this latter section, a discussion of the dietary and health implications of the changes would be relevant. Inclusion of relevant health outcome measures like concurrent trends in anthropometric measures, nutrition-related outcomes, burden of diseases, etc. would be particularly relevant here together with comparisons of the results of intake of nutrition to dietary reference values for Bangladesh, alternatively the WHO recommendations.
A further weakness is in the projection section, again because of the lack of health implications of the results. To a nutrition related audience it would be warranted to add physiological and potential health implications to the statistical findings together with a discussion about the relevance of using the method applied.
Specific points
Line 61: The information that the ‘’later surveys showed inadequacies in calories and about half in proteins is unclear and would benefit from quantification of the caloric and protein intake in relation to reference values.
Line 74-75: Information is provided that since 1973 only households Expenditure Surveys are available from Bangladesh. Maybe the authors are not aware of the dietary surveys referred to in reference 9, which describes two dietary surveys conducted in 1981-82 and 1995-96. The information should be added.
Line 145: Specify that the ‘nutrients’ referred to here are actually macronutrients.
Line 609-611. The sentence ‘the increasing consumption of carbohydrate in the diet leads to decrease in consumption of other protein and fat’ makes little sense since the data clearly show that energy intake has increased.
Line 611. The sentence ‘Bangladesh has made progress in apparent protein and fat intake level’ is unclear. It is not known how the ‘progress’ is evaluated.
Line 615. It is unclear which structural changes that are referred to – and how such changes have increased the dietary diversity. The latter should be clearly defined.
Figures: The text in all figures should be enlarged to allow readers to also be able to read on a screen.
Author Response
Reply to respected reviewer's comment
Title of the study: Temporal trends in apparent energy and macronutrient intakes in the diet in Bangladesh: a joinpoint regression analysis of FAO’s food balance sheet data from 1961 to 2017
Respected reviewer: 1
Date: 16 July 2020
Comment 1: The current manuscripts aims to analyze the temporal trends and changes apparent energy and macronutrient consumption in Bangladesh from 1961 to 2017 and to estimate the projected apparent energy and macronutrient intakes in the diet in Bangladesh unto 2050. The manuscript has a clear structure and a method section with appropriate information to allow replication. The analyses of the dynamics and temporal trends in apparent energy and macronutrient intake in Bangladesh is highly pertinent.
Reply 1: Dear Sir/Madam, we are indebted to you. Thank you very much for your valuable comment and kindness. We are grateful to you for mentioning the strength and importance of our research.
Comment 2: My major consideration and the weakness of the current manuscript in the present form is the lack of nutritional or health implications of the findings. The outcome measures (trends in apparent energy and macronutrient intake) are shown in the results, discussed in relation to historical events in Bangladesh for the various subsections and repeated again in the discussion. Especially in this latter section, a discussion of the dietary and health implications of the changes would be relevant. Inclusion of relevant health outcome measures like concurrent trends in anthropometric measures, nutrition- related outcomes, burden of diseases, etc. would be particularly relevant here together with comparisons of the results of intake of nutrition to dietary reference values for Bangladesh, alternatively the WHO recommendations.
Reply 2: Dear Sir/Madam, thank you very much for pointing out the major pitfalls of our manuscript and we have learned many things from your valuable comment. We have tried our best to provide the concurrent trends of nutritional and health implications of our findings. We agree with you that the previous manuscript had lack in nutritional and health implications of findings. In the revised manuscript we have discussed about the dietary and health implications of the changes. We have included relevant health outcome measures like concurrent trends in anthropometric measures, nutrition-related outcomes, and the burden of diseases that would be particularly relevant with the comparisons of our results. In our revised manuscript, in the following line number we have tried to discuss the nutrition and health implications of our findings.
Revised line no: 412-419 (discussed the nutritional deficiency during the 1960s);
Revised line no: 431-443 (discussed the protein-energy malnutrition and child growth retardation and the prevalence of diabetes and mentioning the level of blood cholesterol and sugar in the population during the 1960s);
Revised line no: 456-469 (dietary intake and deficiency in energy and protein intakes compared to the recommended level mentioned in the literature of that time, also showed the prevalence of malnutrition in under-five children and 5-11 years old children during the 1970s and 1980s);
Revised line no: 496-509 (mentioned about how the increased energy availability due to high rice yields predicted child malnutrition, discussed the changing trends in chronic energy deficiency in women and chronic and acute malnutrition in children);
Revised line no: 559-568 (discussed the trends in over weight in women and children in Bangladesh since the 1990 in relation to the inadequate proportionate balance of macronutrients in the diet);
Revised line no: 577-589 (discussed the deficiency of iron, riboflavin, and protein in the population during the 1960s);
Revised line No: 612-615 (discussed shortly about the protein intake trend and underweight trend in the children);
Revised line no: 665-677 (discussed the global burden of disease report for Bangladesh as a summary of the situation since 1990).
Comment 3: A further weakness is in the projection section, again because of the lack of health implications of the results. To a nutrition related audience it would be warranted to add physiological and potential health implications to the statistical findings together with a discussion about the relevance of using the method applied.
Reply 3: Dear Sir/Madam, thank you very much for your valuable comment. We agree with you that because of the lack of health implications of the results the projection is not scientifically sound. Moreover, Sir/Madam, actually the projection that we did in our analysis is mathematical extrapolations of the values. True to say, I have a lack of knowledge on this section and do not know how to do this analysis in a more scientifically justified way. So we have decided to drop the entire projection section from our study from the title to the conclusion. Here we are mentioning the line numbers regarding the deletion. Revised line no: 2 (Title);
revised line no: 18-19 (Abstract);
revised line no: 31-35 (Abstract);
revised line no: 99-100 (Introduction);
revised line no: 165-175 (Methods);
revised line no: 301-324;
revised line no: 325-331 (Figure 3);
revised line no: 345-353 (Table 3);
revised line no: 392- 398 (Discussion);
revised line no: 634-658 (Discussion);
revised line no: 690-694 (Discussion)
Specific Points
Comment 4:
Line 61: The information that the ‘’later surveys showed inadequacies in calories and about half in proteins is unclear and would benefit from quantification of the caloric and protein intake in relation to reference values.
Line 74-75: Information is provided that since 1973 only households Expenditure Surveys are available from Bangladesh. Maybe the authors are not aware of the dietary surveys referred to in reference 9, which describes two dietary surveys conducted in 1981- 82 and 1995-96. The information should be added.
Line 145: Specify that the ‘nutrients’ referred to here are actually macronutrients.
Reply 4:
Dear Sir/Madam, thank you very much for your valuable comment. Sir/Madam, we agree with you that the statement is unclear. So according to your suggestion we have put the values of intake in different survey years. The reference values of intake in energy, protein and fat were also reported on the literature which we have cited. Revised line no: 61-65 and Revised line No: 69-72
Sir/Madam, thank you very much. Yes, we have missed mentioning the information. In our revised manuscript we have added the information (Revised line no: 69-72). In this case, we have not given information from the reference 9, because this was a study which was conducted in two villages of Bangladesh and that was not a national survey. We have
provided the information (Revised manuscript reference no: 10) of the intake values from the national survey during the same period mentioned in reference 9.
Sir/Madam, we have corrected according to your suggestion (Revised line No: 157)
Comment 5:
Line 609-611. The sentence ‘the increasing consumption of carbohydrate in the diet leads to decrease in consumption of other protein and fat’ makes little sense since the data clearly show that energy intake has increased.
Line 611. The sentence ‘Bangladesh has made progress in apparent protein and fat intake level’ is unclear. It is not known how the ‘progress’ is evaluated.
Line 615. It is unclear which structural changes that are referred to – and how such changes have increased the dietary diversity. The latter should be clearly defined.
Reply 5:
Dear Sir/Madam, thank you very much for your valuable comment. Yes, the statement makes little sense and so we have deleted the sentence in the revised manuscript (Revised line no: 738-739).
We have tried to clear the sentence in the revised manuscript (Revised line no: 739- 741). We want to express that the apparent intakes level of protein and fat have increased and progressed in the sense of intake level. As we have no native English-speaking author so the word choice and use are not clear to some extent. We are sorry for this.
We have tried to make the statement clear in the revised manuscript (Revised line no: 743-746). In the later statement, we have talked about the substitution effect (Revised line no: 749-751) which is one of the components of these structural changes in the diet. Moreover, we have also mention of dietary diversity.
Comment 6: Figures: The text in all figures should be enlarged to allow readers to also be able to read on a screen.
Reply 6: Dear Sir/Madam, thank you very much for your valuable comment. We have increased the font size of the text in all figures (12pt) in the revised manuscript.

Reviewer 2 Report
The authors present energy availability (including macronutrient availability and %) for the Bangladeshi diet from 1961 to 2017. These data come from Food Balance Sheets (FBS). The authors also project availability into the future.
This is a worthwhile analysis that provides new insight into the diet of people in Bangladesh. However, I have concerns regarding data quality and interpretation:
1. The authors acknowledge the limitations of FBS but they do nothing to try and determine whether the data are accurate. Are there other types of data that would corroborate the FBS data? For example, there are two HIES surveys noted in the Figures and mentioned briefly in the text. These surveys give different values for energy and macrontrient availability than the FBS data. Why do these studies disagree? If the disagreement is due to error in the FBS, how can we be sure the trends we see in the FBS are real and not themselves due to error? Perhaps there are other sources of data: average body weights, % stunting, average BMI, GDP, or other dietary data that could be used to compare to the FBS data and see if they are accurate and reliable. There needs to be some investigation of the data and whether they are sound. Otherwise, the reader has little confidence that the trends are real.
2. The authors briefly touch on the social history of Bangladesh, noting that the 1971 liberation war may be the cause of reduced energy availability (although I would note that it could also have caused a change in the recording of FBS data). That's very helpful context, but please give us some similar historical context for the other changes after liberation. The rates of change are detailed several times in the manuscript, but there is no discussion of potential causes or context.
3. The projections to 2050 are simply extrapolations of the last couple years of data. I don't see any reason to expect these changes will occur as suggested. It's possible, of course, but as the authors themselves show, the rate of change in availability changes dramatically over time. Why should we accept the authors' claims that energy availability will change as they predict?
4. It is misleading to state that energy availability has changed very little from 1961 until 2017. If the FBS data are correct, energy availability has changed a great deal. It only happens that the value today, after 57 years of change, has come back to the level seen in 1961. To compute rates of change between 1961 and 2017 values isn't meaningful.
I would encourage the authors to investigate the reliability of the FBS data, examine how and why FBS estimates differ from (or agree with) other indicators of energy availability, provide more context, and drop the projections. The text can also be revised to avoid redundancy and colloquial phrases ("at a snail's pace", etc)
Author Response
Reply to respected reviewer's comment
Title of the study: Temporal trends in apparent energy and macronutrient intakes in the diet in Bangladesh: a joinpoint regression analysis of FAO’s food balance sheet data from 1961 to 2017
Respected reviewer: 2
Date: 14 July 2020
Comment 1: The authors present energy availability (including macronutrient availability and %) for the Bangladeshi diet from 1961 to 2017. These data come from Food Balance Sheets (FBS). The authors also project availability into the future.
Reply 1: Dear Sir/Madam, thank you very much for your valuable comment and kindness. Thank you very much for summarizing our study. Thank you very much also for commenting on the strength of our study.
Comment 2: This is a worthwhile analysis that provides new insight into the diet of people in Bangladesh. However, I have concerns regarding data quality and interpretation:
-
The authors acknowledge the limitations of FBS but they do nothing to try and determine whether the data are accurate. Are there other types of data that would corroborate the FBS data? For example, there are two HIES surveys noted in the Figures and mentioned briefly in the text. These surveys give different values for energy and macrontrient availability than the FBS data. Why do these studies disagree? If the disagreement is due to error in the FBS, how can we be sure the trends we see in the FBS are real and not themselves due to error? Perhaps there are other sources of data: average body weights, % stunting, average BMI, GDP, or other dietary data that could be used to compare to the FBS data and see if they are accurate and reliable. There needs to be some investigation of the data and whether they are sound. Otherwise, the reader has little confidence that the trends are real.
Reply 2: Dear Sir/Madam, thank you very much for your valuable comment. In the following table we have tried to summarize the nutritional dataset that are available in Bangladesh as per our concern.
|
Area |
Available dataset |
Year available |
|
National level |
Food balance sheet |
1961-2017 |
|
|
Bangladesh Bureau of Statistics |
1974-2019 |
|
Household level |
Bangladesh National Nutrition Survey |
1975-76, 1981-82, 1995-96 |
|
|
Household Income expenditure Survey |
1985, 1988, 1991, 1995, 2000, 2050, 2010, 2016 |
|
|
Bangladesh Integrated Household Survey |
2011-12, 2015 |
|
|
Bangladesh Bureau of Statistics |
1974-2019 |
|
Individual level |
Bangladesh Integrated Household Survey |
2011-12, 2015 |
These are the available survey that has been conducted in Bangladesh. At the national level, Bangladesh Bureau of Statistics is the centralized official bureau that has been collecting information on demographics, economy, agriculture, health, and other facts about Bangladesh. The agriculture wing is responsible for producing statistics on the agriculture of Bangladesh. They compile annual estimates of the acreage, production and yield rate of six major crops in Bangladesh. The agriculture wing follows the concepts, the definitions, and the recommendations of FAO.
Our study aimed to analyze the energy and macronutrient availability at the national level of Bangladesh and for this reason; Bangladesh has no other data sources except the FAOSTAT. So for our analysis, we have to rely on only the database provided in the FAOSTAT. There are no other data sources which can be corroborated continuously every year for this long period from 1961 to 2017. The other data source that can be corroborated is Household Income expenditure Survey (HIES) but this survey has different a point of view; this survey does not give information about the food availability or food available for consumption at the national level or apparent food consumption. The HIES surveys provide information about the food access or actual food intake at the household. Where FAOSTAT data (that we have used in our study) provide information about food availability or apparent food consumption; the HIES survey provides information about the food access or actual food consumption at the household level. Hence the information provided by both of the surveys will never be the same. We have mentioned this in the limitation of our study
(Revised line no: 696-701). The difference is not an error but the limitations of informing some issue more accurately.
If we plan to compare the FAOSTAT data with other available databases like HIES survey, or body-weight, stunting, average BMI, or other dietary data then we cannot conduct the pair-wise comparison of the trends as the other data sources are not continuous. They have been collected in a specific time interval and have values at a single point in time. We can only compare on a single point in time but not in a continuous way. A comparison study [1] has already done by using FAOSTAT data and HIES surveys data on seven single point in time (1985, 1988, 1991, 1995, 2000, 2005, and 2010), but not continuously from 1985 to 2010. This study quantified the changes comparing values of specific point in time but did not provide the continuous temporal trend. Moreover, in our revised manuscript we have tried to discuss the available trend in malnutrition status, energy deficiency, stunting, and BMI among the population of Bangladesh.
Humbly, we want to provide some more information about the reliability of the FAOSTAT database that we have used in our study for trend analysis. A review paper [2] of 119 studies that used the FAOSTAT database in their study has reported the reliability of food balance sheets in health research. We have included the justifications in our revised manuscript (Revised line no: 717-721) to overcome the limitations of our data sources.
Comment 3: 2. The authors briefly touch on the social history of Bangladesh, noting that the 1971 liberation war may be the cause of reduced energy availability (although I would note that it could also have caused a change in the recording of FBS data). That's very helpful context, but please give us some similar historical context for the other changes after liberation. The rates of change are detailed several times in the manuscript, but there is no discussion of potential causes or context.
Reply 3: Dear Sir/Madam, thank you very much for your valuable comment. Sir/Madam, I agree with you that the Liberation War could also have caused a change in the recording of FBS data. In our revised manuscript, we have tried to give some of the historical context for the changes after the Liberation War; like Green Revolution in Bangladesh (Revised line no: 476-478), policy issue of the government (Revised line no: 478-481), Expansion of aquaculture (Revised line no: 609-611). Moreover, in our previous analysis, we have provided a detailed context for the changes in apparent food consumption in Bangladesh which we briefly summarized in this revised manuscript.
Comment 4: 3. The projections to 2050 are simply extrapolations of the last couple years of data. I don't see any reason to expect these changes will occur as suggested. It's possible, of course, but as the authors themselves show, the rate of change in availability changes dramatically over time. Why should we accept the authors' claims that energy availability will change as they predict?
Reply 4: Dear Sir/Madam, thank you very much for pointing out the major pitfalls of our analysis. We agree with you that the projection that we did in our analysis is mathematical extrapolations of the values. Sir/Madam, it is not confirmed that the energy availability will change as we predicted because this is merely a mathematical extension of values. Moreover, because of the lack of health implications of the results the projection is not scientifically sound. True to say, I have a lack of knowledge on this section and do not know how to do this analysis in a more scientifically justified way. Thank you very much for your valuable suggestion. We have decided to drop the entire projection section from our study from the title to the conclusion. Here we are mentioning the line numbers regarding the deletion. Revised line no: 2 (Title); revised line no: 18-19 (Abstract); revised line no: 31-35 (Abstract); revised line no: 99-100 (Introduction); revised line no: 165-175 (Methods); revised line no: 301-324; revised line no: 325-331 (Figure 3); revised line no: 345-353 (Table 3); revised line no: 392-398 (Discussion); revised line no: 634-658 (Discussion); revised line no: 690-694 (Discussion).
Comment 5: 4. It is misleading to state that energy availability has changed very little from 1961 until 2017. If the FBS data are correct, energy availability has changed a great deal. It only happens that the value today, after 57 years of change, has come back to the level seen in 1961. To compute rates of change between 1961 and 2017 values isn't meaningful.
Reply 5: Dear Sir/Madam, thank you very much for your valuable comment. Maybe we are wrong in judging the extent of changes in energy availability. Moreover, we are not even sure how to quantify the change. We agreed with your judgment about the change and have corrected the misleading terms in our revised manuscript. The corrections are as follows- Revised line no: 179-180 in the Results section; revised line no: 379 in the Discussion.
Comment 6: I would encourage the authors to investigate the reliability of the FBS data, examine how and why FBS estimates differ from (or agree with) other indicators of energy availability, provide more context, and drop the projections. The text can also be revised to avoid redundancy and colloquial phrases ("at a snail's pace", etc)
Reply 6: Dear Sir/Madam, thank you very much for your valuable comment. Sir/Madam, we have tried to provide some judgment for the reliability of the FBS data (Revised line no: 717- 720) and humbly in the reply of your Comment 1, we have tried to provide some information about the difference. As per your valuable suggestion, we have dropped the projection section completely from our analysis from the title to the conclusion. As we have no native English-speaking author so the word choice and use are not clear to some extent. We are really sorry about this. Hence, we will contact with MDPI for professional English editing service for our manuscript.

Round 2
Reviewer 1 Report
The current manuscripts aims to analyze the temporal trends and changes apparent energy and macronutrient consumption in Bangladesh from 1961 to 2017 and to estimate the projected apparent energy and macronutrient intakes in the diet in Bangladesh unto 2050. The manuscript has a clear structure and a method section with appropriate information to allow replication. The analyses of the dynamics and temporal trends in apparent energy and macronutrient intake in Bangladesh are highly pertinent.
The authors have made valuable improvements in the present revised version. The outcomes of the analyses are discussed in relation to historical distinct periods in Bangladesh and in relation to the nutritional or health implications. That part has improved the reading considerably.
However, there are still some aspects of the paper that need to be considered to improve the readability and value of the present results.
Major points
- The authors have published an earlier paper (ref 7) on similar Food Balance Sheet Data using similar analytical tools, as introduced very briefly and too short in the introduction (line 66). It would be valuable to have a brief description of the major findings in the background for this paper.
- With all the uncertainties inherent in data analyses at this level, it is demed pertinent that the figures given are rounded off to a reasonable level. As an example, in line 617 the text is: ‘about 40.3% of the population” doesn’t make much sense. Please, change this to a more appropriate figure like “around 40%” and do it likewise for the rest of the paper.
- Methodology section: it is unclear on what basis the drivers of the transformation observed in apparent energy and nutrient intake are identified? , describe.
Minor points
Line 84: Pls., provide the recommended level directly in figures.
Line 95: Pls., make a new paragraph with the sentence on Food Balance sheets.
Line 132: Use the abbreviations for FBDG as introduced already.
Line 142-150: These sentences are not about methodology – rat to the introduction – and should be moved.
Line 166: Pls., describe the desirable dietary patterns.
Line 256: unclear what “apparent protein energy is”? is it both (requiring an ‘and’) or is it a ratio between them? Pls., clarify and correct accordingly.
Line 447: choric? Maybe a typo?
Line 546: describe what kind of study, the ref 31 is.
Section 4.1 and 4.2. These sections are very lengthy and would benefit from removal of information that has already been described elsewhere. An example is the information on the calorie deficits in line 804-808 in the section on macronutrients. Please, revise and condense.
Figure 3: pls, return this figure to the original. It seems more logical to have the trend over years described in the X-axis (and similar to the authors’ previous paper (no 7)).
Figures in general: The text in all figures should be enlarged to allow readers to also be able to read on a screen.
Author Response
Reply to respected reviewer's comment
Title of the study: Temporal trends in apparent energy and macronutrient intakes in the diet in Bangladesh: a joinpoint regression analysis of FAO’s food balance sheet data from 1961 to 2017
Respected reviewer: 1 (2nd Round)
Date: 27 July 2020
_______________________________________________________________________
Comment 1: The current manuscripts aims to analyze the temporal trends and changes apparent energy and macronutrient consumption in Bangladesh from 1961 to 2017 and to estimate the projected apparent energy and macronutrient intakes in the diet in Bangladesh unto 2050. The manuscript has a clear structure and a method section with appropriate information to allow replication. The analyses of the dynamics and temporal trends in apparent energy and macronutrient intake in Bangladesh are highly pertinent.
Reply 1: Dear Sir/Madam, we are indebted to you. Thank you very much for your valuable comment and kindness. Your valuable comment and suggestions have improved this manuscript tremendously and able to connect our findings to the nutritional and health implications in Bangladesh. We are grateful to you for mentioning the potency and importance of our research.
Comment 2: The authors have made valuable improvements in the present revised version. The outcomes of the analyses are discussed in relation to historical distinct periods in Bangladesh and in relation to the nutritional or health implications. That part has improved the reading considerably.
Reply 2: Dear Sir/Madam, thank you very much for your kind words. Improvement is solely credited to the valuable suggestions and a different point of view we have got and learned from your suggestions and comments.
Major Points
Comment 3: The authors have published an earlier paper (ref 7) on similar Food Balance Sheet Data using similar analytical tools, as introduced very briefly and too short in the introduction (line 66). It would be valuable to have a brief description of the major findings in the background for this paper.
Reply 3: Dear Sir/Madam, thank you very much for your valuable comment. We agree with you that a brief description of our previous published work can connect the readers with the findings of this paper. Based on your valuable suggestions, in this manuscript we have discussed the nutritional and health implications of our findings which we had not pointed out in our previous work. In our previous work we have discussed about the drivers that have changed the food availability in Bangladesh but in this manuscript beside the drives of change, we have discussed about the health and nutritional implications. A very brief description of the major findings of our previous has added to this revised manuscript in the revised line no: 61-65 (Introduction).
Comment 4: With all the uncertainties inherent in data analyses at this level, it is demed pertinent that the figures given are rounded off to a reasonable level. As an example, in line 617 the text is: ‘about 40.3% of the population” doesn’t make much sense. Please, change this to a more appropriate figure like “around 40%” and do it likewise for the rest of the paper.
Reply 4: Dear Sir/Madam, thank you very much for your valuable comment. True to say, I was not aware of this. I just put the reported values. I agree with you that the figures are better to read and understandable when rounded off to a reasonable level. In the revised manuscript, we have rounded off most of the figures. Revised line no: 57, 71, 72 (Introduction); revised line no: 225, 250, 283, 284, 290, 328, 329, and 331 (Results); revised line no: 424, 477, 508, 512, 513, 514, 515, 550, 551, 552, 560, 561, 590, 608, 609, 610, 611, 620, 632, 633, 634, 643, 644, 661, and 713 (Discussion).
Comment 5: Methodology section: it is unclear on what basis the drivers of the transformation observed in apparent energy and nutrient intake are identified? , describe.
Reply 5: Dear Sir/Madam, thank you very much for your valuable comment. We have taken the joinpoints as the basis when the drivers of the transformation observed. We have discussed this in our revised manuscript in the revised line no: 177-182.
In each joinpoint the trend significantly alters its direction and observes the transformation or changes in energy and macronutrient intakes in the diet. Hence, each joinpoint acts as point of changes in energy and macronutrient intakes and is the basis of the transformation observed. We conceptualized that changes in food availability, technological advancement and policy-driven growth in agriculture, and population growth might have acted as drivers [1-3] for this joinpoints, the node of changes in apparent energy and macronutrient intakes in the diet. In the discussion section we have tried to present some of the possible drivers that acted to changes in the availability of energy and macronutrients at national level. Among the drivers we have discussed the food availability, initiation of agricultural technologies such as improved cultural practices, introduction of improved irrigation facilities, and introduction of high yielding varieties. The secular decline in per capita availability during the 1970s and 1980s was due to the stagnation of production due to the natural calamities and disruption owing to the war of liberation. We have discussed about this issues as the drivers of the stagnation or declining phase of the availability of nutrients at national level. We have discussed about the effect of the policy of the government which affected the availability of food at national level in very briefly (We have discussed more in our previous publication [3]) such as trade liberalization, liberalizing input markets, import, and tariff reduction policy. In our discussion section we have tried our best to discuss about some of the possible drivers of changes. We have not discussed about some of the drivers such as income, economic growth, food prices, urbanization, and effect of transactional food corporations, retailing process, food industry marketing policies, and consumer’s attitude and behaviors. Because all these drivers are directly and with a great extent related to the actual food consumption. These factor works on the accessibility and actual consumption of food [1]. Thank you very much for your valuable suggestion which improves our manuscript a lot. We are indebted to you.
- Kearney, J. Food consumption trends and drivers. Trans. R. Soc. B Biol. Sci.2010, 365, 2793–2807.
- Headey, D.D.; Hoddinott, J. Agriculture, nutrition and the green revolution in Bangladesh. Syst.2016, 149, 122–131.
- Al Hasan, S.M.; Saulam, J.; Kanda, K.; Hirao, T. Temporal Trends in Apparent Food Consumption in Bangladesh: AJoinpoint Regression Analysis of FAO’s Food Balance Sheet Data from 1961 to 2013. Nutrients 2019, 11, 1864.
Minor Points
Comment 6:
Line 84: Pls., provide the recommended level directly in figures.
Line 95: Pls., make a new paragraph with the sentence on Food Balance sheets.
Line 132: Use the abbreviations for FBDG as introduced already
Line 142-150: These sentences are not about methodology – rat to the introduction – and should be moved.
Reply 6:
Dear Sir/Madam, thank you very much for your valuable comment. We have provided the recommended level directly in figures in our revised manuscript in the revised line no: 86-88
Sir/Madam, thank you very much. We have made the new paragraph in the revised manuscript in the revised line no: 103
Sir/Madam, we have used the abbreviation according to your suggestion. Revised line no: 103, 104, and 107 (Introduction); Revised line no: 123, 129, 130, and 137 (Methodology); Revised line no: 753, 757, 759, 763, 768, and 772 (Discussion)
Dear Sir/Madam, thank you very much for your valuable comment. We have removed the sentences from the methodology section in the revised manuscript (Removed revised line no: 140-148).
Comment 7:
Line 166: Pls., describe the desirable dietary patterns.
Line 256: unclear what “apparent protein energy is”? is it both (requiring an ‘and’) or is it a ratio between them? Pls., clarify and correct accordingly.
Line 447: choric? Maybe a typo?
Line 546: describe what kind of study, the ref 31 is.
Reply 7:
Dear Sir/Madam, thank you very much for your valuable comment. We have described the desirable dietary pattern in the revised manuscript (Revised line no: 167-168).
Sir/Madam, by apparent protein energy we want to mean the percentage of available energy from protein. In the text, carbohydrate energy, protein energy, and fat energy is referring to the percentage of available energy from carbohydrate, percentage of available energy from fat, and percentage of available energy from fat, respectively.
I have typed it wrong. It will be chronic; corrected in the revised manuscript (Revised line no: 421).
We have described the type of study which referring to the reference no-31; in the revised manuscript in the revised line no: 507-508.
Comment 8:
Section 4.1 and 4.2. These sections are very lengthy and would benefit from removal of information that has already been described elsewhere. An example is the information on the calorie deficits in line 804-808 in the section on macronutrients. Please, revise and condense.
Figure 3: pls, return this figure to the original. It seems more logical to have the trend over years described in the X-axis (and similar to the authors’ previous paper (no 7)).
Figures in general: The text in all figures should be enlarged to allow readers to also be able to read on a screen.
Reply 8:
Dear Sir/Madam, thank you very much for your valuable comment. Sir/Madam, We agree with you that this section is very lengthy and we have tried to remove some of the repeated things and condense sentences. As we have shortcomings in English language so we may be not that much successful in condensing and removing. We have removed some parts and tried to condense some sentences in the revised manuscript (Removed revised line no: 465-468; Removed revised line no: 500-501; Removed revised line no: 564-566; Removed revised line no: 620-621; Removed revised line no: 671-674; Removed revised line no: 725-730)
Sir/Madam, I am sorry for this as I think about look rather than scientific logic. We agree with you that it is logical to put the year in the X-axis. In the revised manuscript, we have returned the figure as you suggested (Revised line no: 775).
Sir/Madam, we have increased the font size of the text in all figures (14pt) in the revised manuscript so that it can be readable. Sir/Madam, we are indebted to you for your valuable time and comments and suggestions.